# ADVERSARIAL MACHINE UNLEARNING: A STACKELBERG GAME APPROACH

## ABSTRACT

This paper focuses on the challenge of machine unlearning, aiming to remove the influence of specific training data on machine learning models. Traditionally, the development of unlearning algorithms runs parallel with that of membership inference attacks (MIA), a type of privacy threat to determine whether a data instance was used for training. Recognizing this interplay, we propose a game-theoretic framework that integrates the attacks into the design of unlearning algorithms. We model the unlearning problem as a Stackelberg game, introducing a two-player dynamic: a defender striving to unlearn specific training data from a model, and an attacker employing MIAs to detect the traces of the data. Adopting this adversarial perspective allows the utilization of new attack advancements, facilitating the design of unlearning algorithms. Our framework stands out in two ways. First, it takes an adversarial approach and proactively incorporates the attacks into the design of unlearning algorithms. Secondly, it utilizes implicit differentiation to obtain the gradients that limit the attacker's success, thus benefiting the process of unlearning. We present empirical results to validate the effectiveness of the proposed framework and algorithm.

## 1 INTRODUCTION

The enactment of the General Data Protection Regulation (GDPR) by the EU has elevated the importance of deleting user data from machine learning models to a critical level. This process is distinctly more intricate compared to removing data from conventional databases. Erasing the data's imprint from a machine learning model necessitates an approach to negate the data's influence on the model comprehensively while maintaining the utility and accuracy of the model.

Beyond this, establishing the true extent to which data influence has been erased from the model poses a substantial challenge (Song & Mittal, 2021). Numerous methods and metrics have been advanced to validate the thoroughness of data removal, each with varying degrees of reliability and efficacy (Guo et al., 2020; Thudi et al., 2022). Nonetheless, adopting an adversarial perspective presents a novel and potentially more robust methodology. In this approach, the focus shifts to simulating possible attacks aimed at ascertaining the data's presence during the training phase of the model. If, within this adversarial framework, an attacker fails to distinguish whether a data point was part of the training set or merely a typical instance of unseen data, it can be construed that the influence of the data point on the model has been successfully expunged.

We leverage advancements from the burgeoning domain of Membership Inference Attacks (MIA) to simulate an adversary (Shokri et al., 2017), therein framing a Stackelberg game (SG) between a defender, tasked with orchestrating the unlearning process, and an attacker deploying MIA to deduce the membership of data in the model's training set. The key idea is for the defender to adjust the unlearned model by utilizing gradient feedbacks from the attacker's optimization problem, moving the model in a direction that limits the effectiveness of the attack, thus achieving the goal of unlearning. Specifically, we formulate the MIA as a utility-maximizing problem, where the utility measures the remaining influence of a data point in the unlearned model. The defender's loss function is defined as a combination of the degradation of model performance and the attacker's utility. We harness the development from implicit differentiation and design a gradient-based algorithm to solve the game, allowing for the seamless integration of the game into existing end-to-end pipelines (Gould et al., 2016; Amos & Kolter, 2017; Agrawal et al., 2019).

The contributions of the present paper are summarized below

1. We propose to evaluate the effectiveness of an unlearning algorithm from an adversarial perspective, inspiring us to develop a game theory framework that enables utilizing advanced MIAs for enhancing the unlearning process.

2. Additionally, we design a gradient-based solution method to solve the game by leveraging implicit differentiation, making it amenable to end-to-end pipelines.

3. Finally, we support the efficacy of the game and the solution method with empirical results.

## 2 RELATED WORK

The first related thread is machine unlearning, which focuses on removing the influence of a subset of data (referred to as the forget set) from a machine learning model. The unlearning approaches are divided into two classes. The first one is exact unlearning, which involves retraining a model on data excluding the forget set. The second one is approximate unlearning. The ideas behind approximate unlearning are twofold. The first is to track the influence of each training data on the updates to a model's weights, allowing for a rollback during unlearning (Bourtoule et al., 2021; Graves et al., 2021; Chen et al., 2022). The second is designing a loss function to capture the objectives of unlearning (e.g., removing the influence of the forget set while maintaining model utility) and modifying the model weights to minimize the loss (Guo et al., 2020; Golatkar et al., 2020; Izzo et al., 2021; Warnecke et al., 2023; Chundawat et al., 2023; Jia et al., 2023). The method proposed in this paper aligns with the second idea. Specifically, we design a loss function that simulates an attacker who evaluates the effectiveness of unlearning dynamically, enabling more informative updates to the model weights. Besides algorithmic developments, Jagielski et al. (2023) proposes a measure to quantify the forgetting during training; Thudi et al. (2022) take a formal analysis on the definition of approximation unlearning and propose methods to verify exact unlearning. Due to space constraint, it is not feasible to provide a comprehensive review of all related studies. We refer the readers to the survey article by Nguyen et al. (2022) for a more exhaustive discussion.

The second related line is membership inference attacks (MIA). Shokri et al. (2017) introduced MIAs, showing the privacy risks of machine learning models. Subsequently, different attack methods are proposed (Chen et al., 2021; Carlini et al., 2022; Ye et al., 2022; Bertran et al., 2023). On the other hand, Carlini et al. (2022) show that existing criteria to evaluate MIAs are limited in capturing real-world scenarios and propose more practical evaluation metrics. In addition, comprehensive evaluation frameworks and tools are developed (Murakonda & Shokri, 2020; Song & Mittal, 2021). Finally, Nasr et al. (2018) propose a defense mechanism to counter MIAs from an adversarial perspective. Our method shares conceptual similarities with this work, but there are several key differences. Our primary focus is on machine unlearning problems, while their focus is on defending against MIAs. This means that our framework needs to support multiple types of MIAs to provide a comprehensive evaluation of unlearning, including both neural network (NN)-based and non-NN based attacks. However, their framework only supports NN-based attacks. Furthermore, NN-based attacks are generally not suitable for our runtime requirements; indeed, if unlearning takes longer than retraining, we would opt for retraining instead.

## 3 PRELIMINARIES

**Machine Unlearning.** Let $D = \{(x_i, y_i) \mid x_i \in \mathcal{X}, y_i \in \mathcal{Y}\}$ be a labeled dataset, where $\mathcal{X}$ (resp. $\mathcal{Y}$) denote the feature (resp. label) space. The training, validation, and test sets are $D_{tr}$, $D_{val}$, and $D_{te}$, respectively. A machine learning (ML) algorithm is denoted by $\mathcal{A}$, mapping from the joint space of features and labels $\mathcal{X} \times \mathcal{Y}$ to a hypothesis class. We refer to the model trained on the entire training set as the original model, i.e., $\theta_o = \mathcal{A}(D_{tr})$. Let $D_f = \{(x_f^j, y_f^j)\}_{j=1}^q \subseteq D_{tr}$ represent a forget set. The goal of machine unlearning is to remove the influence of the forget set from the original model, resulting in an unlearned model $\theta_u$ (i.e., $\theta_u = \mathcal{U}(\theta_o)$) where $\mathcal{U}$ represents a machine unlearning algorithm. There are typically two settings in machine unlearning that differ in the sampling of the forget set. One is class-wise, where the forget instances $x_f$ are sampled from a single class. The other is randomly sampled, where the instances are uniformly sampled at random from all the

classes. We focus on the latter setting, which is based on real-world experiments that show forget requests rarely come from a single category (Bertram et al., 2019).

The unlearning algorithm may have access to other inputs (e.g., the validation set $D_{val}$) depending on the problem settings. Let $D_r$ be the retain set, the subset of the training data excluding the forget set, i.e., $D_r = D_{tr} \setminus D_f$. The gold standard of machine unlearning is $\theta_r = \mathcal{A}(D_r)$, a model trained on the retain set, excluding the influence of $D_f$. We use $\theta_r$ as a gold standard for comparing machine unlearning algorithms. Retraining is expensive, especially for deep neural networks. This motivates the development of efficient machine unlearning algorithms that satisfy the following conditions: 1) the influence of $D_f$ does not exist in the unlearned model, 2) the performance of the unlearned model is comparable to the performance of the original model, and 3) computational costs (e.g., running time) are cheaper than those of retraining.

**Membership Inference Attacks.** A membership inference attack (MIA) aims to determine whether a data instance was used to train an ML model (Shokri et al., 2017). An instance that is in the training set is called a member, while one that is not in the training set is called a non-member. Formally, given a target model $\theta$, an attacker infers the membership of an instance $(x, y)$ based on the model's outputs (i.e., $\theta(x)$) and the label. The attacker does not have access to either the training data or the model parameters of the target model. Instead, he gathers proxy training and test sets and learns a model $\tilde{\theta}$ to mimic the behavior of the target model. Using the predictions of $\tilde{\theta}$ on its own training and test data, the attacker acquires a labeled (member v.s. non-member) dataset. and then uses the labeled dataset to train a binary classifier for determining the membership of an instance.

We adapt the idea of MIA for determining whether the influence of the forget set still exists in an unlearned model $\theta_u$. Define an auditing set $\tilde{D}_{\theta_u} = \{(s_f^j, 1), (s_{te}^j, 0)\}_{j=1}^q$, where $s_f^j$ (resp. $s_{te}^j$) represents the outputs of the forget (resp. test) instances from the unlearned model, that is, $s_f^j = \theta_u(x_f^j)$. Here, the test instances serve as an empirical distribution for the unseen data. The outputs can be scalars, such as the cross-entropy losses; alternatively, they can be the vectors of probabilities across the classes (Shokri et al., 2017; Carlini et al., 2022). The labels "1" and "0" indicate members and non-members, respectively. The MIA reduces to a binary classification task on $\tilde{D}_{\theta_u}$, aiming to differentiate the forget instances from the test ones based on the outputs.

## 4 SETUPS FOR THE GAME FORMULATION

We model the machine unlearning problem as a Stackelberg game (SG) between a defender who deploys models as services, and an attacker who launches MIAs against the model. The key idea is to assess the effectiveness of an unlearning algorithm by measuring whether the attack succeeds. In particular, the unlearning is considered effective when the attacker is unable to differentiate between the forget set from the test set based on the outputs from the unlearned model. The SG is played in a sequential manner: the defender first deploys an unlearned model, then the attacker launches an MIA in response. Importantly, the advantage of first-mover endows the defender with the power to make a decision knowing that the attacker will play a best-response (i.e., launching a strong attack). We now formally define the models for both players.

### 4.1 THE ATTACKER'S MODEL

We begin by defining the attacker's model. Suppose the defender has deployed an unlearned model $\theta_u$. Following standard setup (Shokri et al., 2017; Song & Mittal, 2021), we assume that the attacker has blackbox access to the model, allowing him to query the model, e.g., submitting data to the model and collecting the outputs. The attacker's goal is to determine whether the influence of the forget set still exists in the model based on the querying outputs. To achieve this, the attacker constructs an auditing set $\tilde{D}_{\theta_u}$, consisting of the model's outputs on the forget and test instances (see Section 3 for details about the auditing set). The attacker assesses the distinctiveness of the two sets with a binary classifier trained on the auditing set through cross validation.

Let $U_a$ be the attacker's utility function, quantifying the distinctiveness of the forget and test instances. Intuitively, a large $U_a$ indicates that the outputs of the forget instances are highly different from the outputs of the test instances, a strong evidence that the influence of the forget set still exists

in the unlearned model. We formulate the attacker's model as the following optimization problem

$$
\begin{aligned}
U_a(\theta_u) &= M(\tilde{D}_{\theta_u}^{val}; \theta_a) \\
\text{where } \theta_a &\in \mathcal{B}(\theta_u) = \arg\max_{\theta_a' \in \mathcal{H}_a} V(\tilde{D}_{\theta_u}^{tr}; \theta_a').
\end{aligned}
\tag{1}
$$

The auditing set $\tilde{D}_{\theta_u}$ is divided into the training set $\tilde{D}_{\theta_u}^{tr}$ and the validation set $\tilde{D}_{\theta_u}^{val}$. The constraint encodes the process of learning a binary classifier; the function $V$ represents the learning objective, e.g., the log-likelihood of the training data. The set $\mathcal{B}(\theta_u)$ are the attacker's best-responses to the defender's decision—the unlearned model $\theta_u$. The function $M$ captures the evaluation of the binary classifier on the validation data. The definition of $M$ is flexible, One can use the accuracy to quantify the average performance of the classifier, where true positives are weighted equally with true negatives (Shokri et al., 2017; Song & Mittal, 2021). Alternatively, an average measure may not capture real privacy threats. Instead, ROC curve or true positive rates at specified false positive rates are employed for the evaluation Carlini et al. (2022).

The attacker's model exhibits a high degree of generality, unifying several advanced MIAs in the literature; this includes neural network-based attacks proposed by Nasr et al. (2018), quantile regression-based attacks from Bertram et al. (2019), and prediction confidence-based attacks by Song & Mittal (2021), etc. Under formulation of (1), the mentioned attacks differ in 1) the hypothesis class $\mathcal{H}_a$ of the binary classifier and 2) the learning objective $V$. Notice the dependence of the attacker's best-response on $\theta_u$ (i.e., $\mathcal{B}(\theta_u)$) arising from the defender's first-mover advantage. The defender utilizes this dependence to select an unlearned model that limits the attacker's utility, which we discuss next.

## 4.2 The Defender's Model

Next, we define the defender's model. Let $C_d$ represent the defender's cost function, which encompasses two main objectives for unlearning. The first objective is to maintain the utility of the model, ensuring that the unlearned model performs comparably (e.g., in terms of predictive power) to the original model. To achieve this objective, we minimize a loss function $L(D_r; \theta_u)$ computed on the retain set $D_r$, following the principles of empirical risk minimization. All regularization terms are included in the loss function to simplify notation. The second objective focuses on eliminating the influence of the forget set from the unlearned model $\theta_u$. We approach this objective adversarially by considering the attacker's utility $M(\tilde{D}_{\theta_u}; \theta_a)$. In essence, a smaller value of the attacker's utility indicates that the forget set is harder to be distinguished from the test set, providing strong evidence that the unlearning process is effective.

Formally, the defender's optimization problem is to minimize the cost function below

$$
C_d(\theta_u) = L(D_r; \theta_u) + \alpha \cdot M(\tilde{D}_{\theta_u}; \theta_a).
\tag{2}
$$

The parameter $\alpha \in \mathbb{R}^+$ balances the loss $L$ and the attacker's utility $M$. Depending on the specific setting, the cost function $C_d$ can be extended to incorporate additional objectives for unlearning. For instance, one can specify that the unlearned model should perform poorly on the forget set (Graves et al., 2021); this can be achieved by minimizing the likelihood of the forget set. Also, several sparsity-promoting techniques have been shown helpful for unlearning (Jia et al., 2023); one way to achieve this is by adding an $\ell_1$ regularization to the cost function.

## 4.3 The Stackelberg Game

Now, with the defender and attacker models in place, we formally define the Stackelberg game (SG). The SG is to solve the following bilevel optimization problem (Colson et al., 2007)

$$
\begin{aligned}
\min_{\theta_u \in \mathcal{H}_d} \quad & L(D_r; \theta_u) + \alpha \cdot M(\tilde{D}_{\theta_u}; \theta_a) \\
s.t. \quad & \theta_a \in \mathcal{B}(\theta_u).
\end{aligned}
\tag{3}
$$

The hierarchical structure encodes the sequential order of the play, with the upper level corresponding to the defender's optimization problem and the lower level capturing the attacker's best-response. During the process of unlearning, the defender needs to proactively consider the attacker's responses.

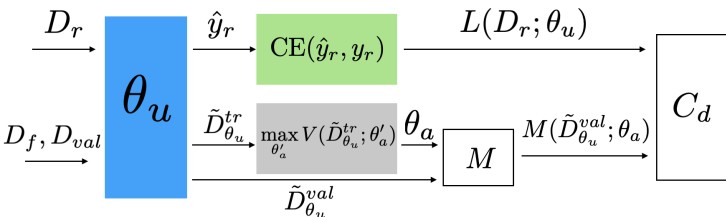

Figure 1: The computational graph of the Stackelberg game. The top path represents a forward passing with a standard loss function (e.g., the Cross-Entropy loss in green). The bottom path represents solving and differentiating the attacker's optimization problem by utilizing Differentiable Optimization. The gradient $\partial C_d / \partial \theta_u$ is obtained by a standard forward-backward pass.

This requires selecting an unlearned model where the influence of the forget set is erased, or from the attacker's perspective, the forget instances are indistinguishable from the test ones.

One assumption of the SG is that if the forget set cannot be distinguished from the test set—in terms of the effectiveness of an MIA—its influence is deemed eliminated from the unlearned model. We justify this assumption from three angles. First, one common way to measure forgetfulness is by assessing the accuracy of the unlearned model on the forget set (Graves et al., 2021; Chundawat et al., 2023; Baumhauer et al., 2022). This approach is based on the observation that machine learning models perform differently on training data compared to unseen data. However, it is important to note that accuracy on the forget set does not necessarily correlate with forgetfulness. This is because there are inherently difficult (or easy) instances that result in low (or high) accuracy regardless of whether they were part of the training set (Carlini et al., 2022). Secondly, MIAs have been used to study training data forgetting (Jagielski et al., 2023), demonstrating its utility in detecting residual traces of a subset of data. Finally, from an adversarial perspective, if a sophisticated attack like an MIA cannot differentiate the forget set from the test set, it is reasonable to expect that the influence of the forget set has been removed.

We solve the SG using gradient-based methods, which allows for easy integration into end-to-end training pipelines. One advantage is that there is no need to solve the attacker's optimization problem separately. Instead, we utilize Implicit Function Theorem to differentiate through the attacker's optimization and compute the gradient $\partial M / \partial \theta_u$. As a result, the SG becomes a differentiable layer, making it compatible with the standard paradigm of forward-backward computation. The solution methods will be detailed in the next section.

## 5 OUR SOLUTION

In this section, we describe the algorithm for solving the Stackelberg game (SG). In general, it is NP-hard to find an optimal solution (Conitzer & Sandholm, 2006). Therefore, our focus is on efficient algorithms to find an approximate solution. The main technical challenge lies in computing the gradient $\partial M / \partial \theta_u$, which requires differentiation through the attacker's optimization problem. This differentiation can be bypassed in some cases, e.g., when the defender's hypothesis class is of linear regressions (Tong et al., 2018). However, this is rarely applicable in our setting as our primary focus is on deep neural networks.

Our solution is to utilize the implicit function theorem (Dontchev et al., 2009) and tools from Differentiable Optimization (DO) to compute the gradient (Gould et al., 2016; Amos & Kolter, 2017; Agrawal et al., 2019). The resulting solution is a gradient-based method, making the SG a differentiable layer that is easily integratable into existing end-to-end pipelines.

We start by expanding the gradient of $C_d$ w.r.t. $\theta_u$ using the chain rule

$$\frac{\partial C_d}{\partial \theta_u} = \frac{\partial L(D_r; \theta_u)}{\partial \theta_u} + \frac{\partial M(\tilde{D}_{\theta_u}^{val}; \theta_a)}{\partial \theta_a} \cdot \frac{\partial \theta_a}{\partial \tilde{D}_{\theta_u}^{tr}} \cdot \frac{\partial \tilde{D}_{\theta_u}^{tr}}{\partial \theta_u}. \tag{4}$$

The gradient $\partial L / \partial \theta_u$ on the right-hand side can be easily computed using an automatic differentiation tool like PyTorch (Paszke et al., 2017). In essence, the computation involves passing the

$D_r$ through $\theta_u$ during the forward pass, computing the loss $L$, and obtaining the gradient in the backward pass. The computational graph is illustrated in Figure 1. The second term on the right is an expansion of $\partial M / \partial \theta_u$ using the chain rule. The gradient $\partial M / \partial \theta_a$ is obtained by performing a standard forward-backward pass. Note that most commonly used metrics (e.g., the 0-1 loss, AUC, recall, etc.) are non-differentiable. Therefore, when needed, we use a standard differentiable proxy for $M$, e.g., using the logistic loss as a proxy for the non-differentiable 0-1 loss.

Computing $\partial \theta_a / \partial \tilde{D}_{\theta_u}$ requires differentiation through the attacker's optimization problem. The main challenge is the absence of an explicit analytical function that maps $\tilde{D}_{\theta_u}$ to $\theta_a$. However, under certain regularity assumptions, one can derive an implicit mapping between $\tilde{D}_{\theta_u}$ and $\theta_a$ based on the optimality condition of the attacker's optimization problem (Gould et al., 2016). A concrete example is when the attacker's optimization problem is convex.[1] In this case, the Karush-Kuhn-Tucker conditions are expressed as a system of linear equations involving $\tilde{D}_{\theta_u}$ and $\theta_a$, i.e.,

$$f(\tilde{D}_{\theta_u}, \theta_a) = 0, \tag{5}$$

where $f$ encapsulates the stationarity conditions, the primal and dual feasibility conditions, and the complementary slackness conditions (Boyd & Vandenberghe, 2004). For illustration purposes, a concrete example of $f$ for linear support vector machines is provided in A.6. We then utilize the implicit function theorem to differentiate (5) and obtain the gradient

$$\frac{\partial \theta_a}{\partial \tilde{D}_{\theta_u}} = - \left( \frac{\partial f(\tilde{D}_{\theta_u}, \theta_a)}{\partial \tilde{D}_{\theta_u}} \right)^{-1} \frac{\partial f(\tilde{D}_{\theta_u}, \theta_a)}{\partial \theta_a}. \tag{6}$$

We refer the readers to the lectures by Gould (2023) for details about differentiating through an optimization problem with the implicit function theorem.

In practice, we capitalize on the tools from Differentiable Optimization (DO) to compute the above gradient. Specifically, we describe the attacker's optimization problem using a certain modeling language, e.g., `cvxpy` (Diamond & Boyd, 2016); the optimization problem is parameterized by $\tilde{D}_{\theta_u}$ and the corresponding optimal solution is $\theta_a$. Then, DO converts the description into a differentiable layer, with the KKT conditions and the implicit differentiation implemented internally. Finally, the differentiable layer is placed on top of the unlearned model $\theta_u$, forming a computational path from $\theta_u$ to $\theta_a$. An illustration of the gradient-based method is shown in Figure 1. In particular, the bottom path shows the process of differentiating through the attacker's optimization problem. The pseudocode is provided in Algorithm 1. The algorithm has time complexity $O(n^3)$, where $n$ represents the size of the attacker's optimization problem (i.e., the number of variables and/or constraints). The cubic dependence results from the matrix inversion in (6).

---

**Algorithm 1** SG-Unlearn

1: Input: $D_r, D_f, D_{te}$ and $\theta_o$
2: Initialize: $i = 0, \theta_u^0 = \theta_o$, a scheduler on $\eta^i$
3: **While** $i <$ **epoch**
4:     $\hat{y}_r^j \leftarrow \theta_u^i(x_r^j)$, where $(x_r^j, y_r^j) \in D_r$
5:     $L(D_r; \theta_u^i) \leftarrow \frac{1}{|D_r|} \sum_j \text{CE}(\hat{y}_r^j, y_r^j)$
6:     $s_f^j \leftarrow \theta_u^i(x_f^j), s_{te}^j \leftarrow \theta_u^i(x_{te}^j)$
7:     $\tilde{D}_{\theta_u^i} \leftarrow \{(s_f^j, 1), (s_{te}^j, 0)\}_{j=1}^q$
8:     `AttOpt` $\leftarrow$ `Model (1) with cvxpy`
9:     `DiffLayer` $\leftarrow$ `DO(AttOpt)`
10:     $\theta_a \leftarrow$ `DiffLayer`$(\tilde{D}_{\theta_u^i})$
11:     $M(\tilde{D}_{\theta_u^i}; \theta_a) \leftarrow$ `Evaluate`
12:     $\frac{\partial L}{\partial \theta_u^i}, \frac{\partial M}{\partial \theta_u^i} \leftarrow$ `Backward`$(M + L)$
13:     $\frac{\partial C_d}{\partial \theta_u^i} \leftarrow \frac{\partial L}{\partial \theta_u^i} + \frac{\partial M}{\partial \theta_u^i}$
14:     $\theta_u^{i+1} \leftarrow \theta_u^i - \eta^i \cdot \frac{\partial C_d}{\partial \theta_u^i}$
15:     $i \leftarrow i + 1$
16: **End**
17: Return: $\theta_u^i$

---

## 6 EXPERIMENTS

### 6.1 EXPERIMENTAL SETUPS

We run experiments on CIFAR-10, CIFAR-100, and SVHN, three well-known image classification datasets (Krizhevsky et al., 2009; Netzer et al., 2011). For all experiments we use the ResNet-18

---

[1]This includes several state-of-the-art MIA (Bertran et al., 2023; Song & Mittal, 2021).

architecture (He et al., 2016). We consider the setting where the forget set is randomly sampled from the training set. For CIFAR-10 and CIFAR-100, the forget set consists of 10% of the training data; for SVHN, the forget set is 5% of the training data. For all experiments, the attacker's optimization problem is instantiated as learning a linear support vector machine (SVM) to classify the forget instances and the test ones.

We compare SG with the following baselines. For all methods, we use the SGD optimizer with a weight decay of 5e-4 and a momentum of 0.9. Other hyperparameters are selected through the validation set. Specifically, we create a new auditing set that includes the outputs of the forget set and the validation set. For each unlearning method, we select the hyperparameters that maximize the difference between the accuracy on the validation set and the MIA accuracy on the new auditing set. Due to space limitations, the hyperparameters are listed in Table 4 in the appendix.

**Retrain:** The first baseline is retraining, where the unlearned model is obtained by training on the retain set from scratch. We aim to develop unlearning algorithms so that the metrics they produce are as closely aligned with those of the retraining as possible.

**Fine-Tuning (FT):** As the second baseline, FT continues to train the original model on the retain set for a few epochs. This a standard baseline used in various prior research (Graves et al., 2021; Warnecke et al., 2023).

**Gradient Ascent (GA):** This baseline takes the original model as the starting point and runs a few epochs of gradient ascent on the forget set $D_f$. The intuition is to disrupt the model's generalizability on $D_f$ (Graves et al., 2021).

**Fisher Forgetting (FF):** As the fourth baseline, FF assumes that the weights of the original model $\theta_o$ are close to those of the retrained model $\theta_r$. Then a step of Newton's method is performed to move $\theta_o$ toward $\theta_r$ (Golatkar et al., 2020).

**Influence Unlearning (IU):** This baseline uses Influence Function to estimate the updates required for a model's weights as a result of removing the forget set from the training data (Izzo et al., 2021; Koh & Liang, 2017).

We evaluate SG and the baselines with the following metrics, which have been adopted in prior studies (Bourtoule et al., 2021; Jagielski et al., 2023; Jia et al., 2023; Chundawat et al., 2023). *It is important to note that the test accuracy is evaluated on a subset of the test data that is separate from the one used for solving SG.*

**Retain accuracy and test accuracy:** These two metrics are used to quantify model utility.

**The accuracy, AUC and F1 score for MIA:** These metrics quantify the effectiveness of unlearning, which are estimated on the auditing set using 10-fold cross validation. A good unlearning algorithm should be close to random guessing in terms of the metrics.

**Forget accuracy:** This measures the accuracy of the unlearned model on the forget set. An effective unlearning algorithm should produce an unlearned model where the forget accuracy is close to the test accuracy. Indeed, if the unlearned model has never seen the forget set, its performance on this set should be consistent with that on the test set.

**The absolute difference between the forget and test accuracy:** This metric measures whether the unlearned model performs consistently on the forget set and the test set.

**Kolmogorov-Smirnov (KS) statistics:** We collect the cross-entropy losses of the forget and test instances from the unlearned model into the empirical distributions $\mathcal{L}_f$ and $\mathcal{L}_{te}$, respectively. Next, we run a KS test to determine if the distributions can be differentiated from each other. The KS statistic quantifies the differences between $\mathcal{L}_f$ and $\mathcal{L}_{te}$; the p-value indicates whether the difference is significant (Massey Jr, 1951).

**Wasserstein distance:** In addition to the KS statistics, we provide the Wasserstein distance between the empirical distributions of $\mathcal{L}_f$ and $\mathcal{L}_{te}$. This complements the KS statistics and evaluates the unlearning performance in terms of the similarity between the losses.

## 6.2 EXPERIMENTAL RESULTS

The experimental results are presented in Table 1. Retraining is considered the gold standard[2], and the results that are closest to the results of retraining are highlighted in bold. SG achieves the best performance for most of the metrics across the three datasets, demonstrating its effectiveness in

---

[2]Notice that for retraining, the KS test cannot differentiate between the forget instances and the test instances based on the cross-entropy losses as indicated by the large p-values.

unlearning. Specifically, the KS statistic of SG is consistently smaller than that of the other baselines, and there is an order of magnitude difference between the statistics on CIFAR-10. Intuitively, ML models behave differently on training data compared to unseen data, and this difference is usually reflected in the corresponding losses (Carlini et al., 2022). The small KS statistic of SG implies that the forget and test instances exhibit greater similarity in terms of the model's behavior, although there is still a discernible difference between the losses. We provide a visualization of the cross-entropy losses for the forget and test instances from one of the experiments in the appendix (Figure 6).

Another observation from the table is that there is a clear trade-off between model performance, measured by test accuracy, and the effectiveness of unlearning, measured by MIA accuracy. Specifically, SG is more effective in unlearning compared to the other baselines. This is shown by the lower values of (MIA) accuracy, AUC, and F1 score. However, this effectiveness comes at a cost to the test accuracy on CIFAR-10 and CIFAR-100, although the degradation is not significant. Indeed, for CIFAR-10, SG experiences on average a 2.8% drop in test accuracy but the MIA accuracy is 6.6% lower (or better) than other baselines; for CIFAR-100, there is on average a 10.8% drop in test accuracy and a 22.7% decrease in MIA accuracy.

Finally, we conduct a comparison study to understand the impact of adversarial modeling on the process of unlearning. In Figure 2, we compare two cases where the trade-off parameter $\alpha$ is set to either 1 or 0, denoted by SG-1 and SG-0 respectively. The comparison is done for four metrics: 1) the test accuracy; 2) the MIA accuracy; 3) the defender's utility, evaluated as the test accuracy minus the MIA accuracy, which provides a combined scalar value that measures both the performance of the unlearned model and the effectiveness of unlearning; 4) the Wasserstein distance between the empirical distributions of $\mathcal{L}_f$ and $\mathcal{L}_{te}$. We show the averages over 10 experiments with different seeds, and 95% confidence intervals are displayed. The first observation is that the adversarial term (i.e., $\alpha \cdot M(\tilde{D}_{\theta_u}; \theta_a)$ in (3)) acts as a regularizer, improving the generalizability of the unlearned model. This observation is supported by comparing the test accuracy of SG-1 and SG-0 on CIFAR-10 (top middle). Similar findings have been reported in Nasr et al. (2018). Another observation is that adversarial modeling limits the attacker's ability to differentiate between the forget set and the test set; this is demonstrated by the MIA accuracy on CIFAR-100 The right-most column displays the Wasserstein distances between $\mathcal{L}_f$ and $\mathcal{L}_{te}$. It is evident that the two losses are considerably closer as a result of adversarial modeling. Additionally, the distances progressively decrease throughout the epochs, confirming the effectiveness of the gradient-based method.

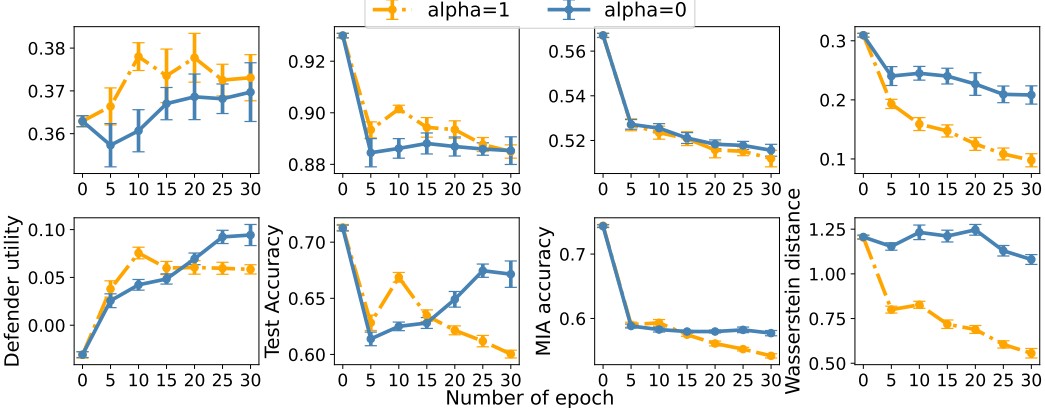

Figure 2: An ablation study to understand the impact of adversarial modeling on the process of unlearning; $\alpha = 1$ and $\alpha = 0$ corresponds to the cases with and without adversarial modeling, respectively. The results are the averages over 10 experiments with different seeds, and 95% confidence intervals are displayed. **From the left to the right**: 1) the defender's utility, evaluated as the test accuracy minus the MIA accuracy; 2) test accuracy; 3) MIA accuracy; 4) Wasserstein distance between the cross-entropy losses of the forget and test instances. **Top**: CIFAR-10; **bottom**: CIFAR-100.

| CIFAR-10 | | | | | | |
|---|---|---|---|---|---|---|
| | Retrain | SG | FT | GA | FF | IU |
| retain acc. | 1 | 0.9953 | **1** | 0.9918 | 0.9997 | 0.9998 |
| test acc. | 0.8805 | **0.8935** | 0.9317 | 0.9159 | 0.9290 | 0.9295 |
| forget acc. | 0.8850 | **0.9249** | 0.9997 | 0.9919 | 0.9997 | 0.9997 |
| \|forget acc. − test acc.\| | 0.0045 | **0.0314** | 0.0685 | 0.0760 | 0.0708 | 0.0702 |
| MIA acc. | 0.5015 | **0.5157** | 0.5610 | 0.5513 | 0.5662 | 0.5662 |
| MIA AUC | 0.5028 | **0.5150** | 0.5927 | 0.5756 | 0.5994 | 0.5998 |
| MIA F1 | 0.5084 | **0.6420** | 0.6929 | 0.6852 | 0.6958 | 0.6960 |
| KS statistic (p-value) | 0.0180 (0.4) | **0.0381** | 0.1608 | 0.1360 | 0.1680 | 0.1682 |
| Wasserstein distance | 0.0435 | **0.1254** | 0.3034 | 0.3321 | 0.3076 | 0.3094 |
| Run time (min.) | 123.46 | 5.73 | 9.82 | 0.71 | 17.81 | 2.74 |
| CIFAR-100 | | | | | | |
| retain acc. | 0.9998 | 0.9947 | **0.9997** | 0.9734 | 0.9996 | 0.9996 |
| test acc. | 0.6209 | **0.6687** | 0.7227 | 0.6726 | 0.7115 | 0.7101 |
| forget acc. | 0.6363 | **0.8595** | 0.9993 | 0.9726 | 0.9998 | 0.9998 |
| \|forget acc. − test acc.\| | 0.0154 | **0.1908** | 0.2766 | 0.2999 | 0.2883 | 0.2897 |
| MIA acc. | 0.5102 | **0.5932** | 0.7344 | 0.6774 | 0.7454 | 0.7452 |
| MIA AUC | 0.5064 | **0.5984** | 0.7544 | 0.7243 | 0.7710 | 0.7716 |
| MIA F1 | 0.5677 | **0.6669** | 0.7890 | 0.7464 | 0.7959 | 0.7959 |
| KS statistic (p-value) | 0.0223 (0.26) | **0.1945** | 0.5156 | 0.4144 | 0.5174 | 0.5193 |
| Wasserstein distance | 0.0703 | **0.8269** | 1.1357 | 1.2950 | 1.2049 | 1.2063 |
| Run time (min.) | 123.03 | 22.60 | 9.30 | 1.43 | 108.34 | 2.83 |

Table 1: Experimental results on CIFAR-10, CIFAR-100. The highlighted metrics are the closest to those of retraining, which is considered as the best performance compared with the other baselines. We provide the p-values for the KS statistics where the differences are *not* significant, meaning that the distributions of looses cannot be differentiated from each other.

## 7 DISCUSSION

In this paper, we design a Stackelberg game framework for addressing the problem of unlearning a subset of data from a machine learning model. Our approach focuses on evaluating the effectiveness of unlearning from an adversarial perspective, conducting membership inference attacks (MIAs) to detect any residual traces of the data within the model. The framework allows for a proactive design of the unlearning algorithm, synthesizing two lines of research—machine unlearning and MIAs—that have heretofore progressed in parallel. By utilizing implicit differentiation techniques, we develop a gradient-based algorithm for solving the game, making the framework easily integrable into existing end-to-end learning pipelines. We present empirical results to support the efficacy of the framework and the algorithm.

The current framework assumes the presence of one defender and one attacker. It is based on the assumption that the attacker uses a single hypothesis class, such as SVM, as we did in the experiments. To expand the current framework, one possibility is to consider a scenario with multiple types of attackers. For instance, the defender may aim to make the unlearned model robust against different types of attacks simultaneously, such as quantile regression-based attacks (Bertran et al., 2023), prediction confidence-based attacks (Song & Mittal, 2021), and NN-based attacks (Nasr et al., 2018). One approach to implementing this scenario is to continue using the current SG framework but regularly update the attack methods. However, this approach may be time-consuming as it requires solving the SG multiple times with different attacks. Additionally, the gradient feedback obtained from these various attacks may no longer provide informative insights.

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

# A APPENDIX

## A.1 EXPERIMENTAL RESULTS ON TEXT DATA AND TRANSFORMER ARCHITECTURES

To verify the effectiveness of the SG framework beyond image data, we conducted experiments using the 20 Newsgroups dataset and a variant of the transformer architecture known as RoBERTa (Liu et al., 2019). We use a pre-trained checkpoint downloaded from HuggingFace. The experimental setup is similar, with the exception that we randomly sample 20% of the training set uniformly to create the forget set. The results are shown in Table 2 and Figure 3.

| 20 Newsgroups | | | |
|---|---|---|---|
| | SG | FT | GA |
| retain acc. | 1.0 | 0.9999 | 0.9129 |
| test acc. | 1.0 | 0.9988 | 0.9349 |
| forget acc. | 1.0 | 0.9990 | 0.9071 |
| \|forget acc. − test acc.\| | 0.0 | 0.0003 | 0.0277 |
| Run time (min.) | 15.6 | 20.2 | 26.1 |
| MIA acc. | 0.5065 | 0.5147 | **0.5057** |
| MIA AUC | **0.4922** | 0.5090 | 0.5080 |
| MIA F1 | **0.5627** | 0.6566 | 0.1579 |
| KS statistic | 0.0791 | **0.0676** | 0.0620 |
| Wasserstein distance | **0.0007** | 0.0100 | 0.1486 |

Table 2: Experimental results on 20 Newsgroups dataset on a pretrained RoBERTa architecture. The gold standard of retraining is too time-consuming to be obtained. For MIA accuracy, AUC and F1, we highlight the metrics that are closest to 0.5—the metrics of random guessing. We highlight the lowest number for KS statistic and Wasserstein distance.

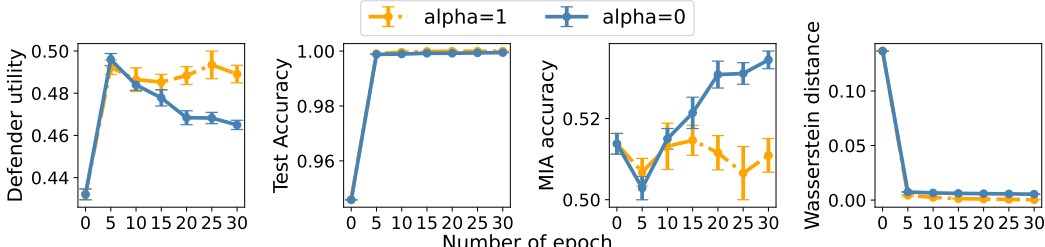

Figure 3: Similar to Figure 2, we conduct an ablation study to understand the impact of adversarial modeling on the process of unlearning; $\alpha = 1$ and $\alpha = 0$ corresponds to the cases with and without adversarial modeling, respectively. The results are the averages over 10 experiments with different seeds, and 95% confidence intervals are displayed.

## A.2 Experimental results on the SVHN dataset

| | SVHN | | | | | |
|---|---|---|---|---|---|---|
| retain acc. | 0.9820 | 0.9396 | 0.9518 | 0.9585 | 0.9584 | **0.9588** |
| test acc. | 0.9540 | **0.9352** | 0.9339 | 0.9333 | 0.9333 | 0.9330 |
| forget acc. | 0.9449 | 0.9278 | **0.9352** | 0.9594 | 0.9604 | 0.9604 |
| \|forget acc. − test acc.\| | 0.0091 | **0.0074** | 0.0013 | 0.0282 | 0.0270 | 0.0304 |
| MIA acc. | 0.5045 | **0.5037** | 0.5014 | 0.5159 | 0.5144 | 0.5142 |
| MIA AUC | 0.4933 | **0.4848** | 0.5120 | 0.5410 | 0.5399 | 0.5397 |
| MIA F1 | 0.1393 | **0.2444** | 0.4704 | 0.6504 | 0.6487 | 0.6491 |
| KS statistic (p-value) | 0.0399 (0.07) | 0.0575 | **0.0519** | 0.0637 | 0.0671 | 0.0669 |
| Wasserstein distance | 0.0422 | **0.0307** | 0.0193 | 0.1029 | 0.0978 | 0.0977 |
| Run time (min.) | 95.10 | 8.73 | 4.78 | 0.52 | 17.53 | 1.89 |

Table 3: Experimental results on SVHN. The highlighted metrics are the closest to those of retraining, which is considered as the best performance compared with the other baselines. We provide the p-values for the KS statistics where the differences are *not* significant, meaning that the distributions of looses cannot be differentiated from each other.

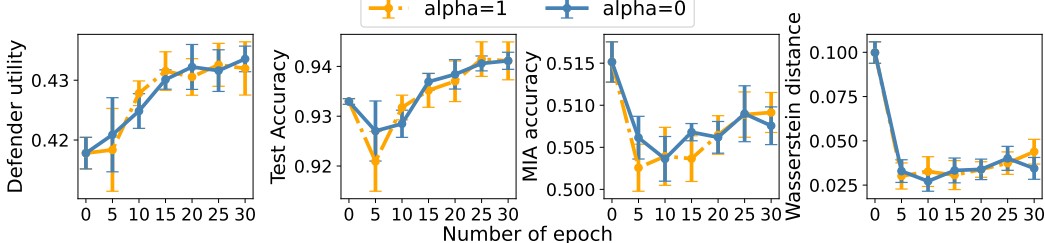

Figure 4: An ablation study to understand the impact of adversarial modeling on the process of unlearning; $\alpha = 1$ and $\alpha = 0$ corresponds to the cases with and without adversarial modeling, respectively. The results are the averages over 10 experiments with different seeds, and 95% confidence intervals are displayed. **From the left to the right**: 1) the defender's utility, evaluated as the test accuracy minus the MIA accuracy; 2) test accuracy; 3) MIA accuracy; 4) Wasserstein distance between the cross-entropy losses of the forget and test instances.

### A.3 THE EFFECT OF THE TRADE-OFF PARAMETER $\alpha$

To gain a better understanding of the trade-off between model performance and unlearning, we conduct experiments using 7 different values for the parameter $\alpha$ (see Eq. (2)), specifically $\alpha \in \{0.05, 0.1, 0.25, 0.5, 1, 2, 5\}$. The results are shown in Figure 5, where each dot represents a batch of 5 random experiments at epoch 10 for a particular value of $\alpha$. The coordinates represent the corresponding metrics averaged across the batch. The red dashed lines denote the metrics for retraining. Ideally, we want the dots to be close to the intersections of the dashed lines. The left column displays the test accuracy compared to the MIA accuracy across the values of $\alpha$. It is evident that the trade-off parameters have varying effects on the unlearning process across different datasets, although the trend is not consistent. The general trend is that a larger $\alpha$ (e.g., $\alpha = 5$) is desirable as it brings the unlearned model closer to the retrained one.

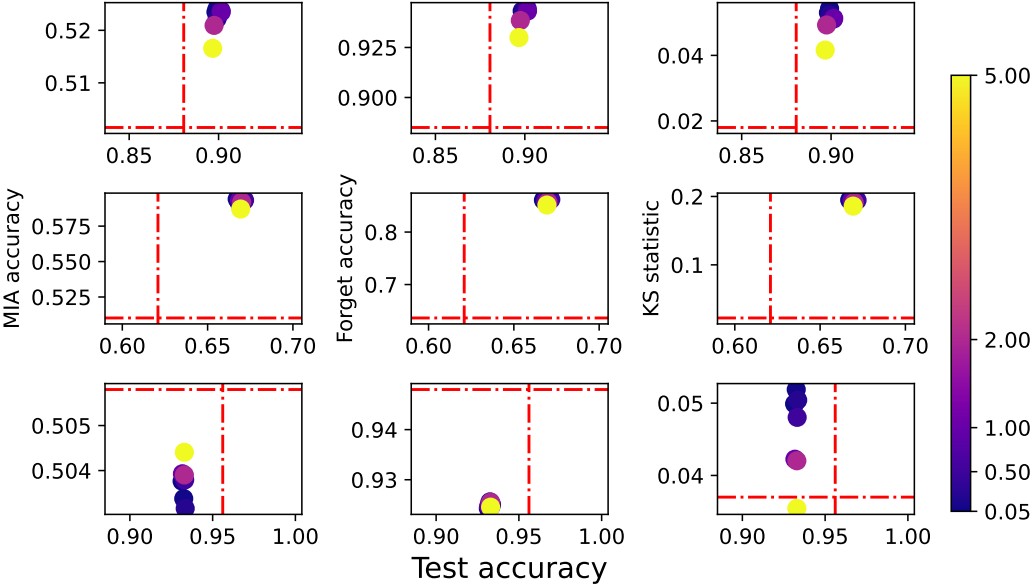

Figure 5: Experiments on different values of the trade-off parameter $\alpha$. We consider 7 values $\{0.05, 0.1, 0.25, 0.5, 1, 2, 5\}$. Each dot represents a batch of 5 random experiments with the same $\alpha$. The coordinates of a dot are the corresponding metrics averaged over the 5 runs. The dashed lines are the performance of retraining, which is considered the gold standards. The colorbar indicates the values of $\alpha$. **Top**: CIFAR-10; **middle**: CIFAR-100; **bottom**: SVHN.

## A.4 BASELINE HYPERPARAMETERS

Tee hyperparameters used for SG and the baselines are in Table 4.

| | Retrain | SG | | FT | GA | FF | IU |
|---|---|---|---|---|---|---|---|
| **CIFAR-10** | | | | | | | |
| learning rate | 0.1 | 0.01 | | 0.01 | 1e-5 | × | × |
| epochs | 200 | 20 | | 5 | 5 | × | × |
| noise level | × | × | | × | × | 1e-9 | 1e-9 |
| **CIFAR-100** | | | | | | | |
| learning rate | 0.1 | 0.01 | | 0.01 | 1e-5 | × | × |
| epochs | 200 | 10 | | 5 | 10 | × | × |
| noise level | × | × | | × | × | 1e-9 | 1e-9 |
| **SVHN** | | | | | | | |
| learning rate | 0.1 | 0.01 | | 0.01 | 1e-6 | × | × |
| epochs | 100 | 15 | | 15 | 15 | × | × |
| noise level | × | × | | × | × | 1e-9 | 1e-9 |
| **20 Newsgroups** | | | | | | | |
| learning rate | × | 1e-5/1e-7 (attacker/defender) | | 1e-5 | 1e-6 | × | × |
| epochs | × | 15 | | 10 | 5 | × | × |
| noise level | × | × | | × | × | × | × |

Table 4: The hyperparameters for all the unlearning methods, which are selected through the validation set. The symbol × indicates that the hyperparameter does not apply to the corresponding method.

## A.5 DISTRIBUTION OF LOSSES

A visualization of the cross-entropy losses of the forget and test instances is in Figure 6.

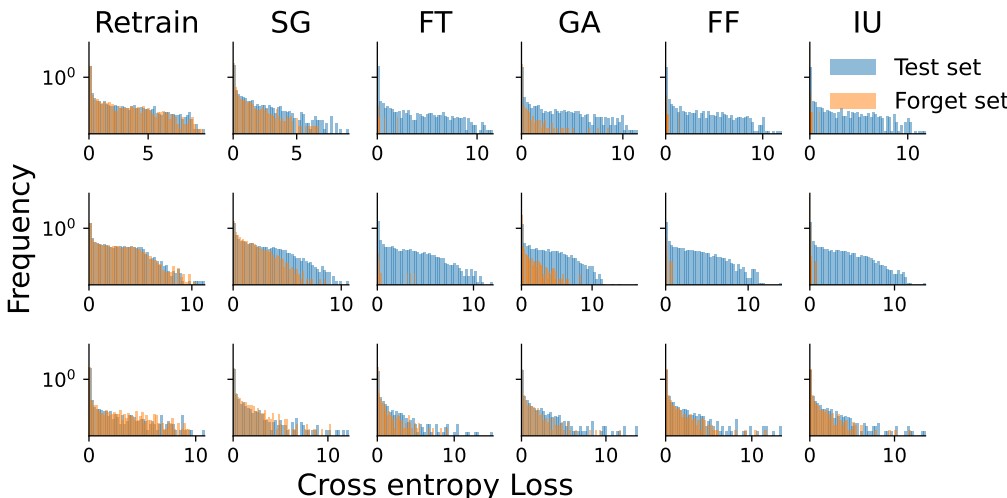

Figure 6: The distributions of the cross-entropy losses for the forget and test instances from the unlearned models. The y-axis is in log scale for better visualization. The columns from left to right correspond to Retrain, SG, FT, GA, FF, and IU. **Top row**: CIFAR-10; **middle row**: CIFAR-100; **bottom row**: SVHN.

## A.6 AN EXAMPLE OF THE CONDITION IN (5)

In this section, we provide a concrete example of the KKT conditions for linear support vector machines (SVM). As described in Section 1, the KKT conditions are key to relating the attacker's model parameters, denoted as $\theta_a$, with the auditing set $\tilde{D}_{\theta_u}$, which allows us to derive the gradient $\partial \theta_a / \partial \tilde{D}_{\theta_u}$. The conditions $f$ can be similarly derived for any model where the learning problem is convex. To simplify the notations, we use $\{(x_i, y_i)\}_{i=1}^q$ to represent $\tilde{D}_{\theta_u}$. A standard formulation of the linear SVM is as follows

$$
\begin{aligned}
\min_{\theta_a, b} \quad & \frac{1}{2}\|\theta_a\|^2 \\
s.t. \quad & y_i \cdot (\theta_a^\top x_i + b) \geq 1, \forall i,
\end{aligned}
\tag{7}
$$

where $b$ is the bias term. The standard form is typically formulated as a minimization problem, so the attacker is to maximize $V = -\frac{1}{2}\|\theta_a\|^2$. Eq. (7) is a convex program, and the optimal solution (i.e., $\theta_a^*$ and $b^*$) is characterized by the KKT conditions. The Lagrangian of the above is as follows where $\alpha_i \geq 0$ are the Lagrantian multipliers:

$$
L(\theta_a, b, \alpha_i) = \frac{1}{2}\|\theta_a\|^2 - \sum_{i=1}^q \alpha_i \left( y_i \cdot (\theta_a^\top x_i + b) - 1 \right).
\tag{8}
$$

Following sandard procedures (Boyd & Vandenberghe, 2004), the KKT conditions are as folllows

$$
f(\tilde{D}_{\theta_u}, \theta_a) = \begin{cases}
\theta_a - \sum_{i=1}^q \alpha_i y_i x_i = 0 \\
-\sum_{i=1}^q \alpha_i y_i = 0 \\
y_i \cdot (\theta_a^\top x_i + b) \geq 1 \\
\alpha_i \geq 0, \forall i \\
\alpha_i (y_i(\theta_a^\top x_i + b) - 1) = 0, \forall i
\end{cases},
\tag{9}
$$

which implicitly define a function between $\theta_a$ and the data $\tilde{D}_{\theta_u} = \{(x_i, y_i)\}_{i=1}^q$. In practice, we describe the optimization problem (7) using `cvxpy` (Diamond & Boyd, 2016). Then, we employ an off-the-shelf package called `cvxpylayers` (Agrawal et al., 2019) to automatically derive the KKT conditions and compute the gradient $\partial \theta_a / \partial \tilde{D}_{\theta_u}$.

## A.7    THE TRAINING LOSS FOR THE RETRAINING BASELINE

The losses for the retraining baseline across the epochs are displayed in Figure 7.

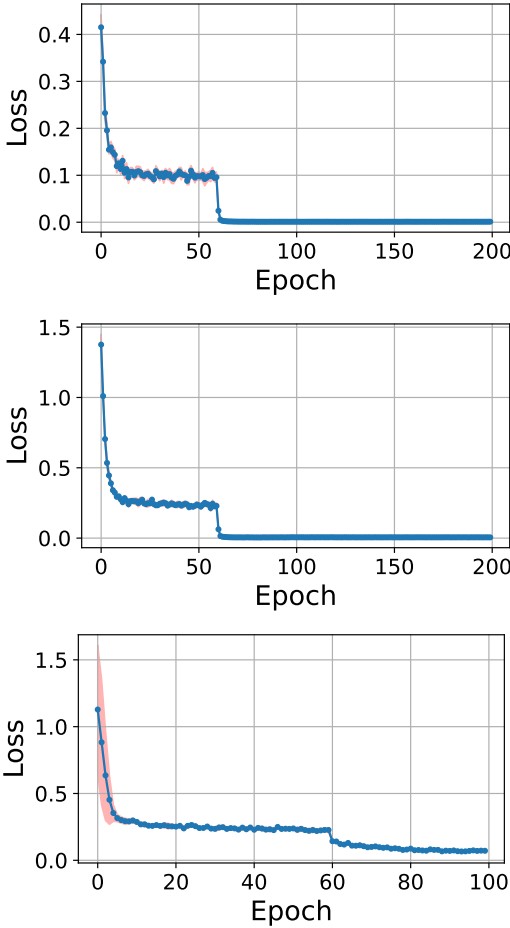

Figure 7:   The training loss for the retrain baseline. For CIFAR10 and CIFAR100, the learning rate is multiplied by 0.1 when epoch is at 60, 120, 160; for SVHN, the same multiplication is done at epoch 60, 120. **Top to bottom**: CIFAR10, CIFAR100, SVHN.

