# OpenReview forum: "Adversarial Machine Unlearning: A Stackelberg Game Approach"
_ICLR.cc/2024/Conference — Submitted to ICLR 2024_

### Official Review · Reviewer_BoXP · 2023-10-23

**Soundness:** 2 fair
**Presentation:** 2 fair
**Contribution:** 3 good
**Rating:** 5
**Confidence:** 4

**Summary:**

This paper introduces a novel machine unlearning method, Stackelberg game (SG) unlearning, which integrates membership inference attack (MIA) into the unlearning algorithms. Specifically, the MIA strives to distinguish forget samples from test samples using outputs generated by the unlearned model. Conversely, the primary objective of the unlearned model is to confound the MIA's attempts to make such distinctions. Then, implicit function theorem is leveraged to solve the SG unlearning with gradient-based solution that fits into end-to-end pipelines. Experiments on CIFAR10 and CIFAR100 show the effectiveness of the proposed SG unlearning method.

The main contribution is that this work proposes to integrate MIA into the design of unlearning, offers a refreshing adversarial perspective to the realm of machine unlearning.

**Strengths:**

- The proposed method SG unlearning is intriguing and novel from my perspective. While the idea of integrating attacks into a defensive framework is not original, this method offers a fresh angle for machine unlearning.
- Addressing the unlearning problem is crucial.

**Weaknesses:**

- Details of the proposed method is not clear.
    - Regarding Eq.(5), what is the precise formulation for $f$? Do the stationarity conditions account fo $\nabla V(\tilde{D}_{\theta_u}^{tr}; \theta_a^{'})$? It would enhance comprehension if the complete equation for $f$ were explicitly presented.

- The manuscript could benefit from more meticulous notation for clarity.
    - In Eq.(4), should $D_{\theta_u}$ actually be $D_{\theta_u}^{val}$? Alternatively, is it meant to be $\tilde{D}_{\theta_u}^{tr}$?
    - In Section 5, when referencing $n$ as indicative of size, is this referring to the size of the attack model parameters?
    - For example, in Section 3, $i$ in $(x_i, y_i)$ denotes the indices, it might be more clear to use $(x_j^f, x_j^f)$ rather than $(x_f^j, x_f^j)$.

- Missing details in the experiments and unconvincing results.
    - All experiments only consider the setting where the forget set is randomly selected, accounting for 10% of the training data, and only employs ResNet-18. A broader set of conditions (e.g., 20% forget set, class-specific forgets, poisoned sample forgets, and varying network architectures) would have enriched the analysis.
    -  What number of epochs is used for the retrained models? And what additional computational overhead does the MIA introduce? It would have been better if these could be provided to show the efficiency.
    - How does the unscrubbed model perform? Also, results in tables for method like FF and IU fail to unlearn, how these methods perform with increased noise level?
    - Considering the MIA is already integrated into the unlearning algorithm, KS and MIA metrics is quite related to the performance of MIA. Incorporating common metrics like Weight Distance [1], Activation Distance [2-3], Relearn Time [4] and Epistemic Uncertainty [5] might offer a more holistic performance view.
    - Besides, it would be better if the loss/accuracy curve (from the retrained model and the proposed method) could be provided to show the convergence.
    - The literature review appears to overlook some recent developments in the field (e.g., [5-8]). It would strengthen the paper's comparative analysis if at least one contemporary method were included for benchmarking.
    - Minor: FT seems achieve the best test acc. in tables.

[1] Tarun, Ayush K., et al. "Fast yet effective machine unlearning." IEEE Transactions on Neural Networks and Learning Systems (2023).

[2] Chundawat, Vikram S., et al. "Can bad teaching induce forgetting? unlearning in deep networks using an incompetent teacher." Proceedings of the AAAI Conference on Artificial Intelligence. Vol. 37. No. 6. 2023.

[3] Golatkar, Aditya, et al. "Mixed-privacy forgetting in deep networks." Proceedings of the IEEE/CVF conference on computer vision and pattern recognition. 2021.

[4] Cao, Yinzhi, and Junfeng Yang. "Towards making systems forget with machine unlearning." 2015 IEEE symposium on security and privacy. IEEE, 2015.

[5] Becker, Alexander, and Thomas Liebig. "Evaluating machine unlearning via epistemic uncertainty." arXiv preprint arXiv:2208.10836 (2022).

[6] Ye, Jingwen, et al. "Learning with recoverable forgetting." European Conference on Computer Vision. Cham: Springer Nature Switzerland, 2022.

[7] Wu, Ga, Masoud Hashemi, and Christopher Srinivasa. "Puma: Performance unchanged model augmentation for training data removal." Proceedings of the AAAI Conference on Artificial Intelligence. Vol. 36. No. 8. 2022.

[8] Chen, Min, et al. "Boundary Unlearning: Rapid Forgetting of Deep Networks via Shifting the Decision Boundary." Proceedings of the IEEE/CVF Conference on Computer Vision and Pattern Recognition. 2023.

**Questions:**

Please see the weakness section.

---

> ### Author Response · Authors · 2023-11-18
> **To reviewer BoXP**
>
> We thank the reviewer for the feedback!
>
> 1.  The precise formulation of $f$ depends on the specific model used in the MIA. The stationarity condition is indeed a part of $f$. For illustrative purposes, we provide an example of $f$ for linear SVM in the appendix (A.6).
>
> 2.1  We thank the reviewer for pointing this out. We have updated Eq.(4) in the revised paper.
>
> 2.2 In section 5, $n$ is the number of variables  plus the number of constraints in the attacker’s optimization problem.
>
> 2.3 We will double check the notations and improve the clarity.
>
> 3.1 We selected 10% to ensure consistent comparison with prior studies. However, we acknowledge that exploring the impact of the forget set size on unlearning is an important direction for future research. To show that the SG framework is applicable to more than just image datasets and ResNet architectures, we present supplementary results on text data and transformer architectures in section A.1 of the appendix.
>
> 3.2 For CIFAR10 and CIFAR100, the retraining process takes 200 epochs. For SVHN, it takes 100 epochs. The major overhead of adversarial modeling consists of two parts: 1) solving the attacker's optimization problem and 2) implicit differentiation. The second part is the most computationally expensive. However, the running time of SG is only a small fraction of the running time of retraining (see Table 1). The detailed hyperparameters for SG and the baselines are now included in Table 4 of the revised paper.
>
> 3.3  The performance of the unscrubbed models (e.g.,  test accuracy and the MIA accuracy) are displayed in Figure 2 at epoch = 0. The noise level of FF and IU is selected on the validation data. The criterion is to select the noise level that maximizes: $\text{prediction accuracy}−\text{MIA accuracy}$. We exhaustively search five values for the noise level: 1e-9, 1e-8, 1e-7, 1e-6, 1e-5. IU is not sensitive to the noise level. However, the test accuracy of FF significantly drops below 0.4 when the noise level exceeds 1e-7, which makes the unlearned model useless for prediction tasks.
>
> 3.4 We would like to clarify that the KS and MIA metrics reported in Table 1 are evaluated on an auditing dataset that is separate from the one used in unlearning. Although the two auditing datasets share the same forget set, they have different test instances. In the revised paper, we have made this distinction clearer. Additionally, we have included two new additions: 1) running time comparisons and 2) the Wasserstein distance between the cross-entropy losses of the forget and test instances. The Wasserstein distance reported in Table 1 is computed on an auditing dataset that is disjoint from the one used in unlearning.
>
> 3.5 We added the training loss for the retrain baseline to Figure 7 in the appendix.
>
> 3.6  We will go through the literature and discuss the connections to the current paper in the next revision.
>
> 3.7 Please refer to comment 6 in “To all reviewers” at the top. We appreciate the reviewer's question and welcome further discussion on this topic.

---

> > ### Comment · Reviewer_BoXP · 2023-11-21
> > **Response to authors**
> >
> > Thank you for the detailed response.
> >
> > However, I still have the following concerns that need to be properly addressed:
> >
> > **1. Current reported results are still not convincing.** For example, SG indeed has the smallest accuracy of the forget set, but its accuracy on the retain/test set is smaller than other methods'. It would be better if the authors could provide the results that prove SG achieves the best trade-off between forgetting and retaining model utility (e.g., SG has larger accuracy than other methods on the test/retain set and smaller accuracy than others on the forget set). Further, it would be better to present the unscrubbed models' accuracy on the retain/test/forget set.
> >
> > **2. Wassertein Distance.** Instead of computing the distance between the forget set and the test set, it would be better to compute the Wassertein distance between the output distribution of the retrain models and that of the scrubbed models [1].
> >
> >  **3.  It is still not clear about the convergence.**  I found Reviewer d9Wv and Reviewer 4Z9k's views on this reconcile with my concern.
> >
> > **4. How would the algorithm perform when forgetting an entire class?**
> >
> > >[1] Tarun, Ayush Kumar, et al. "Deep regression unlearning." International Conference on Machine Learning. PMLR, 2023.
> >
> > Please highlight changes for ease of tracking.

---

> ### Author Response · Authors · 2023-11-21
> **Additional responses to the reviewer's comments**
>
> Thanks for the reviewer’s comments. We highly appreciate the interactive discussion.
>
> 1.1 We argue that a reasonable metric to validate the claim that "SG achieves the best trade-off between forgetting and retaining model utility" is to measure the closeness of SG to the retrained model in terms of various metrics, such as test accuracy, retain accuracy, MIA accuracy, etc. **Specifically, we want to emphasize that the accuracy on the forget set is a misleading metric. If the forget set has a significantly lower or higher accuracy compared with typical testing data, it would be easily identified as abnormal by an attacker---hence breaching privacy**.  We believe that the metrics highlighted in Tables 1 and 3 support the effectiveness of SG. These tables demonstrate that SG is the closest to the retrained model from various perspectives.
>
> 1.2 The asked metrics for the unscrubbed model is below. Each number is averaged over 10 random seeds.
> |               | test (95% conf.)     | retain (95% conf.) | forget (95% conf.) |
> |---------------|----------------------|--------------------|--------------------|
> | CIFAR-10      | 0.92994 (0.000869)   | 0.999782 (0.000014)| 0.99976 (0.000127) |
> | CIFAR-100     | 0.71256 (0.002596)   | 0.999644 (0.000011)| 0.999800 (0.000098)|
> | 20 Newsgroups | 0.945831 (0)         | 0.915061 (0)       | 0.909136 (0)       |
> | SVHN          | 0.932952 (0.000575)  | 0.959251 (0.000456)| 0.959692 (0.001827)|
>
>
> 2. We are calculating the Wasserstein distance between the output distributions of the forget and test sets. This distance provides direct evidence that the two sets are difficult to distinguish. We acknowledge that the Wass. distance you suggested could provide further evidence, and we will include it in the next revision.
>
> 3. We provided the theoretical evidence of the convergence in comment 5 in “To all reviewers”. An exact convergence to a Stackelberg equilibrium would be time-consuming. Instead, the number of epochs of SG is considered a hyperparameter that balances the runtime, the performance of unlearning, and the utility of the model. We hope this resolves the concerns. If not, it would be appreciated if you could expand on your concerns.
>
>
>
> 4. Conceptually, there is no obstacle to generalizing the SG framework to the scenario of forgetting an entire class. However, one challenge is defining what it means to forget a class.  Additionally, as we argued above, the accuracy on the forget set (or 1 - the accuracy on the forget set) would not be an appropriate measure. We acknowledge this as a potential area for future research.

---

> > ### Comment · Reviewer_BoXP · 2023-11-21
> >
> > Thanks again for the clarification. I am still not convinced of the results.
> >
> > * **Class-wise unlearning and Accuracy on the forget set** is a common setting and metric used in machine unlearning papers (e.g., [1-5]). All presented metrics are used for measuring whether the scrubbed model is close to the retrained model. Also, if perform class-wise unlearning [2-3] or forget poison samples [5], the accuracy on the forget set would be quite different from that on the test/retain set. The scrubbed model then is supposed to behave similar to the retrained model on these subsets.
> >
> > * **Theoretical evidence for convergence** seems is not provided, the authors have comments 'Building on their results (Proposition 8), the gradient updates converge to a local Stackelberg equilibrium given an appropriate initialization and mild assumptions. We will add a subsection to discuss the convergence of the gradient-based method.' I didn’t find dicussion in the revised manuscript. **Wasserstein distance** used in [1] for further verification and **Concerns about the clarity** are also not addressed in the revised manuscript.
> >
> > - - -
> > >[1] Tarun, Ayush Kumar, et al. "Deep regression unlearning." International Conference on Machine Learning. PMLR, 2023.
> > >
> > >[2] Jia, Jinghan, et al. "Model Sparsity Can Simplify Machine Unlearning." Thirty-seventh Conference on Neural Information Processing Systems. 2023.
> > >
> > >[3] Golatkar, Aditya, Alessandro Achille, and Stefano Soatto. "Eternal sunshine of the spotless net: Selective forgetting in deep networks." Proceedings of the IEEE/CVF Conference on Computer Vision and Pattern Recognition. 2020.
> > >
> > >[4] Nguyen, Thanh Tam, et al. "A survey of machine unlearning." arXiv preprint arXiv:2209.02299 (2022).
> > >
> > >[5] Sommer, David Marco, et al. "Towards probabilistic verification of machine unlearning." arXiv preprint arXiv:2003.04247 (2020).

---

> ### Author Response · Authors · 2023-11-21
> **Using the accuracy on the forget set as an evaluation metric**
>
> Thanks for the comments.
>
> 1. We agree that studying unlearning in a class-wise setting is an important problem and we leave this as a future research. However, we respectively disagree the use of **smaller** accuracy on the forget set as an evaluation metric for our setting, i.e., the forget set is sampled uniformly at random.  Let's examine the following thought experiment:
>
>     - Imagine you have an unlearned model with a known testing accuracy of 0.9. Now, if you are given a forget set D and the model performs at 0.1 on D, would you suspect that this set has been tampered with by a third-party? If not, how to explain the accuracy gap (0.9 vs 0.1)?
>
>
>
> 2. **Theoretical evidence for convergence**: The discussion has not been added yet. We are currently working on it.
>
>
> 3. ~~**Wasserstein distance used in [1]**: We will consider adding this result.~~ See the new responses below.
>
> 4. **Concerns about clarity**: We will update the notations to better clarify if this is what you are referring to.

---

> > ### Author Response · Authors · 2023-11-22
> > **Wasserstein distance of the output cross-entropy losses**
> >
> > As suggested by the reviewer, we calculated the Wasserstein distance between the cross-entropy losses of the retrained model and each baseline. The specific calculation is as follows
> > $$
> > \frac{\left( \text{Wasserstein distance}(L_\text{forget}^{\text{retrain}}, L_\text{forget}^{\text{baseline}}) + \text{Wasserstein distance}(L_\text{test}^{\text{retrain}}, L_\text{test}^{\text{baseline}}) \right)}{2},
> > $$
> > where $L$ denotes the vector of the cross-entropy losses on the corresponding set. The results below do not include the 20 Newsgroup dataset because retraining the underlying RoBERTa model is too time-consuming. SG is closer to the retrained model in terms of the W-distances. The only exception is SVHN; however, the difference is not statistically significant.
> >
> >
> > |                     | SG     | FT | GA | FF | IU |
> > |---------------|----------------------|--------------------|--------------------|--------------------|--------------------|
> > | CIFAR-10      | **0.272239 (0.002304)** | 0.351157 (0.004369           | 0.307880 (0.004262) | 0.311517 (0.004169) | 0.310659 (0.004224) |
> > | CIFAR-100     | **0.676271 (0.004139)** | 1.028799 (0.005892)          | 0.874748 (0.005823) | 0.884977 (0.005719) | 0.888156 (0.005752)|
> > | SVHN          | 0.097228 (0.005514)          | **0.088863 (0.003368)** | 0.100234 (0.002697) | 0.101083 (0.002710) | 0.100514 (0.002704)|

---

> > > ### Comment · Reviewer_BoXP · 2023-11-22
> > >
> > > Thanks for providing the results. I have updated my score to 5, given that the response partially addressed my concern.
> > >
> > > * To evaluate the unlearning effect, the accuracy of the forget set is one of the metrics to measure (close to a retrained model). Regarding the example the author mentioned, if the training data contains poison samples $D_p$, after using unlearning algorithms to remove the influence of $D_p$, the scrubbed model could then have low accuracy on $D_p$. Perform unlearning algorithms in this case to protect the model's functionality.
> > >
> > > * Its hard to conclude which method is better based on the current reported results in the paper. For example, FT on SVHN has the closest forget accuracy and the smallest MIA score.
> > >
> > > * It would be better if the author could provide results in a class-wise setting. Also, how the proposed method perform compared with [1]?
> > >
> > > ---
> > > > [1] Jia, Jinghan, et al. "Model Sparsity Can Simplify Machine Unlearning." Thirty-seventh Conference on Neural Information Processing Systems. 2023.

---

> ### Author Response · Authors · 2023-11-22
> **Response to reviewer BoXP**
>
> We thank the reviewer for updating the score and we truly appreciate the discussion!
> Our responses are listed below:
>
> 1. We totally agree that removing the influence of poison samples is a crucial aspect of unlearning. The current paper serves as an initial step in adopting an adversarial perspective on machine unlearning. Further investigations into poison samples will be left as a future task.
>
> 2. For SVHN, we conducted two student's t-tests on the MIA accuracy and the forget accuracy (between SG and FT), respectively. The difference in MIA accuracy was not found to be statistically significant (p-value=0.1921). However, the difference in forget accuracy was found to be significant (p-value=0.0003). Despite this, we would argue that SG is not expected to outperform the others in every metric; indeed, it is only explicitly optimized for test accuracy and MIA accuracy (see Eq.(3)).
>
> 3. Class-wise forgetting: we thank for the advice. For having a solid and complete evaluation, we will leave this setting as a future task.
>
> 4. Comparison with [1]: We believe that SG is orthogonal to the method proposed in [1]. Specifically, the method in [1] requires a pruning step, which necessitates access to the weights of the model to be unlearned. However, SG does not require this access. Instead, SG can be seen as a specially designed loss function with an adversarial component. In conclusion, these two methods approach unlearning from different angles and with different assumptions (i.e., white-box v.s. black-box access to the model weights). Therefore, a fair comparison between the two would not be appropriate.

---

### Official Review · Reviewer_4Z9k · 2023-10-30

**Soundness:** 3 good
**Presentation:** 3 good
**Contribution:** 3 good
**Rating:** 5
**Confidence:** 4

**Summary:**

This paper proposes a novel framework for machine unlearning, which aims to remove the influence of specific training data from machine learning models。

**Strengths:**

•	It introduces a game-theoretic approach that integrates membership inference attacks (MIAs) into the design of unlearning algorithms, enhancing the robustness and reliability of unlearning.
•	It develops a gradient-based algorithm that uses implicit differentiation and differentiable optimization to solve the game, allowing for easy integration into end-to-end pipelines.
•	It validates the effectiveness of the framework and algorithm on two image classification datasets, showing a balanced trade-off between model utility and unlearning effectiveness.

**Weaknesses:**

•	It does not provide any theoretical analysis of the game, such as convergence, complexity, or optimality guarantees or discussions.
•	It does not consider multiple attackers or multiple attack methods, which may pose stronger or more stealthy threats to the unlearning process.
•	It does not conduct experiments on more datasets or more complex models, to demonstrate the generalization and scalability of the framework and algorithm.

**Questions:**

Please address the weaknesses above.

---

> ### Author Response · Authors · 2023-11-18
> **To reviewer 4Z9k**
>
> We thank the reviewer’s comments for improving the paper!
>
> 1. Please refer to comment 5 in “To all reviewers” at the top. We will add a subsection to discuss the convergence in details.
>
> 2. We have a time complexity analysis of Algorithm 1 in section 5, just above the pseudocode of the algorithm. We will revise the writing to emphasize the analysis.
>
> 3. We acknowledge the significance of investigating the setting of multiple attack methods as an important direction for future research. It is worth noting that the single-attack setting has not been explored in the context of machine unlearning, and this paper takes a first step towards addressing the unlearning problem from a game theoretic perspective.
>
> 4. We have added additional experimental results on the SVHN dataset, and results on text data and transformer architectures.
>   - SVHN dataset: see A.2 in the appendix.
>   - Text data and transformer architectures: see A.1 in the appendix.
>   - Running time comparison has been added to Table 1.

---

### Official Review · Reviewer_d9Wv · 2023-11-07

**Soundness:** 3 good
**Presentation:** 3 good
**Contribution:** 3 good
**Rating:** 6
**Confidence:** 4

**Summary:**

In summary, the paper presents a novel approach to machine unlearning by framing it as a Stackelberg game where a defender aims to remove specific training data from a model while an attacker uses MIAs to detect traces of the data. This approach combines the fields of machine unlearning and MIAs and leverages gradient-based techniques to enhance the unlearning process, ultimately providing a more robust and effective solution.

**Strengths:**

The proposal of a Stackelberg game framework for addressing the problem of machine unlearning is a notable originality. Using game theory to model the interaction between the defender and attacker in the unlearning process adds a new dimension to this research area. This framework allows for a systematic and strategic approach to unlearning.

The use of implicit differentiation to design a gradient-based solution method for the game is another novel idea. This enables more efficient and effective optimization of unlearning, making it amenable to end-to-end pipelines.

**Weaknesses:**

1. The authors need to further clarify the selection of metrics and justify how they can benefit real-world applications. This should goes back to the objective of machine unlearning. The objective of machine unlearning is to to negate a subset of data’s influence on the model. The goal should be maintaining a high performance of the model while erasing the imprint of the data from the model. Hence, if I understand it correctly, the performance should be as high as possible regardless of the retraining performance as long as the effectiveness of the unlearning is acceptable.

2. The authors consider the setting where the forget set is randomly sampled. First, It is not clear to me why this assumption will hold in real-world scenario when the machine unlearning is motivated by regulations in certain geographical regions. Second, if the forget set is randomly sampled, from a statistical point of view, the problem of machine unlearning becomes the problem of understanding how training dataset size can affect the model performance (the difference between the original performance and the retraining performance).

3. In the empirical results, the authors only demonstrated the effectiveness of the framework. The missing piece is the effectiveness of the proposed gradient-based method. It seems that the paper is lack of demonstration on which point the gradient-based method is converging to. For example, how the utilities of both players evolve during the training process? And which solution concept the algorithm is converging to if it is converging?

**Questions:**

My questions are left in the "Weaknesses" section.

---

> ### Author Response · Authors · 2023-11-18
> **To reviewer d9Wv**
>
> We thank the reviewer’s insightful comments!
> 1. Please refer to comment 6 in “To all reviewers” at the top. We appreciate the reviewer's question and welcome further discussion on this topic.
>
> 2. Uniform sampling is used as an approximation of the distribution of real-world forget requests. For instance, [2] demonstrates that requests to delete URLs originate from multiple countries, although some countries have more requests than others. We acknowledge that studying a more realistic distribution of the forget set is an important direction for future research. Another reason we use the uniform sampling is for having a consistent comparison with prior studies.
> - [2] Five years of the right to be forgotten, Bertram, et al.
>
> 3. We acknowledge the importance of understanding the impact of the training set size on unlearning. However, we believe that the issue of unlearning extends beyond the size of the training data. For example, SG and FT are trained using the same retain set, but the attacker's perspective reveals significant differences in their MIA accuracy and KS (as shown in Table 1). Furthermore, we have included Figure 2 in the revised paper, which demonstrates that adversarial modeling has a significant impact on the learning dynamics. Please refer to 1c & 1d in "To all reviewers" at the top for further details.
>
> 4.  Please refer to comment 1b and 5 in “To all reviewers” at the top (also Figure 2 in the revised paper). We will add a subsection to discuss the convergence in details.

---

### Author Response · Authors · 2023-11-18
**To all reviewers**

We would like to express our gratitude to all the reviewers for their valuable feedback. Based on the feedback, we have revised the paper and included several new empirical results. The revisions are summarized below.
1.  Updated Figure 2:
 - 1a. The figure includes a detailed comparison between the cases with and without the attacker, providing us a better understanding of the impact of adversarial modeling on the unlearning process.
 - 1b. The utility of the defender and the utility of the attacker (e.g., MIA accuracy) are plotted across the epochs. The defender's utility initially increases and then levels off, while the attacker's utility experiences a significant drop at the beginning and then gradually decreases. This shows the effectiveness of the gradient-based method.
 - 1c. The right-most column shows a significant difference in Wasserstein distance between the cross-entropy losses of the forget and test instances, showing another evidence to support the effectiveness of the adversarial modeling.
 - 1d. Overall, even with the same training set, we observe that adversarial modeling brings benefits to unlearning from two aspects. Firstly, it helps in finding an unlearned model that generalizes better (i.e., higher test accuracy). Secondly, it assists in disguising the forget data to the test ones (i.e., lower MIA accuracy and Wasserstein distance).

2.  New experimental results on text data and transformer architectures (i.e., RoBERTa) are provided in section A.1 in the appendix.

3. Table 1 has been updated with a running time comparison. Additionally, the Wasserstein distance between the cross-entropy losses of the forget and test instances is provided.

4. Table 4 in the appendix details the hyperparameters for SG and the baselines. All the hyperparameters are selected on the validation data that maximizes the criterion: prediction accuracy−MIA accuracy.

5. Convergence of the gradient-based method: The dynamics of our gradient updating follow the Stackelberg learning dynamics studied in [1]. Building on their results (Proposition 8), the gradient updates converge to a local Stackelberg equilibrium given an appropriate initialization and mild assumptions. We will add a subsection to discuss the convergence of the gradient-based method.
  - [1] Convergence of Learning Dynamics in Stackelberg Games, Tanner Fiez, Benjamin Chasnov, Lillian J. Ratliff

6. SG is inferior to some baselines on test accuracy: We acknowledge that an unlearned model that outperforms the alternatives across all metrics would be desirable, if such a "Pareto optimal" model existed. However, even the widely accepted gold standard of retraining is found to be inferior to FT in terms of test accuracy Therefore, we argue that  the objective of unlearning is twofold: 1) to make the forget instances indistinguishable from the test instances, and 2) to maintain reasonable predictive power. The former is evaluated using metrics such as MIA accuracy (or AUC, F1 score, etc.), while the latter is compared against the retrained model—a model that exists and can be readily obtained as a reference.

---

### Meta-Review · Area_Chair_qrpr · 2023-12-18

**Metareview:**

An interesting paper but currently underpar for several reasons:

1- uniform sampling: the paper claims to tackle "the challenge of machine unlearning", citing the GDPR as an example in the first introductory sentence. In this framework, uniform sampling is not applicable, not relevant and in fact misleading. Either the paper should have made a clear statement that it is going to revolve on a setting which is lightweight compared to the ones that could eventually be covered in practice, *or* it should remove the uniform sampling assumption.

2- convergence: after the reviewers request, the authors only promise that they *will* add a section on this, further adding it requires appropriate initialization and "mild" assumption. Given the state of the assumptions on which the work is already based (see 1-), it would have been morethan fair to provide details on those additional assumptions, initialization, etc. .

3- Importance of the training set size: after d9Wv's question, the authors essentially say that it is important and then somehow dodge the issue. Given the *formulation* of the machine unlearning problem, it makes no mystery that the size of the training sample is in fact crucial and should not be avoided.

4- literature: to the numerous references pointed by BoXP, the only reply by the authors is that they *will* discuss this in a revision. Why not discussing it at rebuttal time ?

**Justification For Why Not Higher Score:**

restricted analysis, sluggish rebuttal.

**Justification For Why Not Lower Score:**

N/A

---

### Decision · Program_Chairs · 2024-01-16

Reject